# A bit or a lot on the side? Observational study of the energy content of starters, sides and desserts in major UK restaurant chains

Magdalena Muc [ORCID], Andrew Jones, Carl Roberts, Florence Sheen, Ashleigh Haynes, Eric Robinson

Department of Psychological Sciences, Institute of Psychology, Health and Society, University of Liverpool, Liverpool, UK

**Correspondence to**
Dr Magdalena Muc;
mmuc@liverpool.ac.uk

## ABSTRACT

**Objectives** Our objective was to examine the kilocalorie (kcal) content of starters, sides and desserts served in major UK restaurant chains, comparing the kcal content of these dishes in fast-food and full-service restaurants.

**Design** Observational study.

**Setting** Menu and nutritional information provided online by major UK restaurant chains.

**Method** During October to November 2018, we accessed websites of restaurant chains with 50 or more outlets in the UK. Menu items that constituted starters, sides or desserts were identified and their kcal content was extracted. Accompanying beverages were not included. We used multilevel modelling to examine whether mean kcal content of dishes differed in fast-food versus full-service restaurants.

**Main outcome measures** The mean kcal content of dishes and the proportion of dishes exceeding public health recommendations for energy content in a main meal (>600 kcal).

**Results** A total of 1009 dishes (212 starters, 318 sides and 479 desserts) from 27 restaurant chains (21 full-service, 6 fast-food) were included. The mean kcal content of eligible dishes was 488.0 (SE=15.6) for starters, 397.5 (SE=14.9) for sides and 430.6 (SE=11.5) for desserts. The percentage of dishes exceeding 600 kcal was 26.4% for starters, 21.7% for sides and 20.5% for desserts. Compared with fast-food chains, desserts offered at full-service restaurants were on average more calorific and were significantly more likely to exceed 600 kcal.

**Conclusions** The average energy content of sides, starters and desserts sold in major UK restaurants is high. One in four starters and one in five sides and desserts in UK chain restaurants exceed the recommended energy intake for an entire meal.

## INTRODUCTION

Overweight and obesity are now common in most of the developed world. For example, in the UK, two in three adults and one in three children are now classed as having overweight or obesity.[1] Although obesity is a multifactorial disease, it is clear that changes to the food environment have been a key factor driving the global obesity epidemic.[2 3]

### Strengths and limitations of this study

► This is the first study of which we are aware to assess the energy content of starters, sides and desserts in the UK eating out sector.

► Our findings will be of use to future evaluations of how the out of home food sector respond to voluntary or mandatory public health actions through food product reformulation.

► Smaller chains and independent restaurants were not included; however, studies indicate that chain and non-chain restaurants tend to serve highly calorific foods.

► We could only use the nutrition data that restaurants made available, which excluded several dishes from our analyses.

Eating out of the home is becoming increasingly common, with 39% of adults reporting eating out at least once a week in a recent UK study.[4] Eating out of the home is associated with higher energy consumption and research suggests that frequently eating out of the home is a risk factor for obesity.[5] The consumption of 'fast-food' meals has been widely identified as a cause for concern, due to the low nutritional quality and high energy content of meals served in these restaurants.[6] Because of this, the out-of-home food sector has now been identified as an area for public health policy intervention in the USA[7] and the UK government is currently considering similar policy action.[8] However, most of the research on the nutritional quality of food eaten out of the home has been conducted in North America, a region with a particularly high prevalence of obesity.[6 9 10] There has been little research examining the nutritional quality of food sold out of the home in the UK, although a small study of meals sold in independent small-scale takeaway outlets has shown that energy content can be excessive.[11]

In a recent study, we examined the kilocalorie (kcal) content of main meals sold by major restaurant chains in the UK.[12] We found that the average kcal content of main meals was high and very few meals adhered to public health recommendations for main meal kcal consumption (≤600 kcal) recently suggested by Public Health England.[13]

Moreover, we found that main meals sold by full-service restaurants tended to be more calorific than those sold by fast-food restaurants, which is consistent with data from North American restaurants.[14] However, previous research has focused on main meals and consumers eating out can be offered a choice of starters, sides and/or desserts on restaurant menus. The aim of the present study was to assess the average energy content of starters, sides and desserts sold in major UK restaurant chains. Based on Robinson et al,[12] we also examined how common it was for starters, sides and desserts to exceed the amount of calories recommended for an entire meal and whether these dishes were more calorific at full service, as opposed to fast-food restaurants.

## METHODS

This is an observational study of the energy content of menu items across large chain restaurants in the UK. We preregistered the study protocol and analysis plan on the Open Science Framework (https://osf.io/6cfdb/).

### Patient and public involvement

No patients or public were involved in this study.

*Restaurant sampling.* Previously,[12] we identified restaurant chains with ≥50 outlets in the UK by consulting market reports listing restaurants with the largest number of UK outlets, and market research ranking UK restaurant chains by annual turnover, popularity, number of users and numbers of outlets.[15–17] If the number of UK outlets was not provided on a restaurant website, this information was requested by email.

### Characterising restaurant types

As in Robinson et al,[12] we classified restaurant chains as fast-food or 'full-service' restaurants using the following definition of fast-food restaurants: *'Restaurants that primarily provide consumers with largely pre-prepared 'quick' meals with little or no table service, with in-store seating and in which take-away orders are likely to account for a significant proportion of orders'.* We classified full-service chains as restaurants where consumers primarily order and are served while seated at a table.[18] Therefore, coffee shops and take-away only outlets were not considered eligible. Previously, two researchers independently categorised each of the included restaurant as fast-food or full-service with any disagreements resolved by a third researcher.[12]

### Data sources

To access current menus and nutritional information, two researchers visited the restaurants' UK webpages during October and November 2018 and accessed online versions of current menus. If a specific geographical location was required to access a restaurant chain menu, we chose London (largest city in the UK) and the first listed location. If a restaurant only had a downloadable menu (PDF), and no website menu, we used the former. If there were several menus (eg, specials menus), only the 'main menu' was used for coding. If there was no menu clearly labelled as the 'main' menu, then we used the restaurant's 'evening menu'.

### Starters, sides, desserts menu options

We examined the kcal content of starters, sides and dessert menu options. We defined a starter/side/dessert item as being a menu option that is not a main meal dish, is an individually sold food item and can be ordered on its own, as opposed to a more specific addition to a menu item (eg, steak sauce, ice cream toppings). We excluded menu items that could not be ordered by all consumers (eg, items from senior citizens menu section, children's menu section) and excluded platters and sharers (unless the menu indicated the number of people per serving) as we could not confidently identify what combination of sharing menu options would constitute a starter, side or dessert for one person. Small plates (tapas) were not eligible unless they were part of a section of the menu that was labelled as starters, sides or desserts. We also excluded menu items with unspecified portion size, such as 'unlimited' or 'bottomless' options as we would not be able to calculate energy content. In instances in which a menu option could be customised at the request of the patron for an additional charge (eg, add extra toppings), we only extracted the default composition of the menu option. In instances in which a menu option required a customer to make an explicit choice (eg, choice of topping for a starter or dessert accompaniment), we identified all possible configurations for the item and recorded each as an individual menu item (eg, chocolate cake with ice cream, chocolate cake with custard). If a menu item appeared on the menu as served with a drink of choice, we excluded it as a scoping exercise indicated that this was uncommon and our focus is on energy content of food items. Finally, to minimise effects of season, we only included options that were available all year round and sold everyday (eg, we excluded dishes sold only on specific days, such as 'soup of the day').

Two researchers independently identified menu options from each restaurant and a third researcher checked their eligibility according to the protocol and resolved any discrepancies (October 2018). If menu sections were not specifically labelled as starters, sides or desserts, researchers categorised individual menu items according to the menu section they would typically be found under in UK restaurants. If there was a disagreement between two researchers, a third researcher made the final decision. As there was a very high number of menu sections that were not eligible (eg, main meals, sharers, drinks, children's menu, Sunday menu), researchers did not record

a classification (eligible vs not eligible) for every item on each menu. As in Robinson *et al*,[12] we used an approximated intercoder consistency by calculating the number of menu items deemed eligible by both researchers versus the number of menu items included by only one of the researchers.

Because variability in menu item kcal content between restaurant types may be in part explained by the two types of restaurant serving different types of dishes, we examined whether there were dishes that were routinely sold by both types of restaurant (eg, side of fries/chips, salad) and compared the average number of kcals for these dishes by restaurant type. Because the names of the same dishes could vary between menus, coding of these items was completed by one researcher and crosschecked by a second researcher.

### Extraction of dish kcal content

Two researchers accessed the online nutritional information for each restaurant (November 2018) and extracted the number of kcals per menu item. A third researcher independently crosschecked kcal extraction for accuracy.

### Statistical analysis

#### Primary analyses—average number of kcals

Menu items were nested within individual restaurants so we planned to use multilevel analyses (levels: menu item, restaurant) with random intercept at the restaurant level and fixed slopes. We first examined if a multilevel analysis was appropriate for starters, sides and dessert kcals separately by examining the portioning of variance attributed to differences in kcals between restaurants (between restaurant variance/(between restaurant variance +within restaurant variance)). We examined the multilevel model fit by comparing the loglikelihood ratio statistic (loglikelihood of the multilevel model—loglikelihood of the single-level model) to a $\chi^2$ distribution with 1 degree of freedom. We used bootstrapping (500 samples) to improve the accuracy of parameter values and reduce bias in parameter estimates. Statistical significance (p<0.05) indicated meaningful variation in kcals of menu items between restaurants and a multilevel model was used. In all statistical tests, $\alpha$ was set at 0.05 and we report 95% CIs for significance testing. Where multilevel modelling was not appropriate, we used conventional frequentist statistics, maintaining p<0.05 as the level of statistical significance.

#### Secondary analyses

Public Health England recommends that adults do not exceed 600 kcal for a complete meal at lunch and dinner.[13] There are no specific recommendations for individual components (eg, energy from sides) of a meal, so we examined how common it was for starters, sides and desserts to be excessive in kcal content by calculating the proportion of menu items that exceed an entire meal's worth of kcals (600 kcal). We examined differences between the two restaurant types (fast food vs full service)

by using multilevel binary logistic regressions when appropriate.

## RESULTS

### Restaurants

Fifty-two eligible restaurant chains were identified and of these 30 restaurants had available menus and nutritional information. We requested this from the remaining chains but only one provided this information. Because we examined main meal accompaniments (starters, sides, desserts), we excluded four restaurants that only tended to sell individual food items that customers choose from to form a meal (eg, pieces of chicken, pieces of sushi) leaving 27 restaurants in the final sample (n=6 fast-food, n=21 full-service restaurants). See table 1 for restaurants included.

### Menu items

Of all the menu items identified by either of two coders (1494), the first coder identified 74 items which were not identified by the second coder (95.1% agreement) and the second coder identified 35 items not identified by the first coder (97.7% agreement), indicating reasonable consistency between the two coders in identifying eligible dishes. The items in disagreement were then reviewed by a third researcher and after these discrepancies were resolved, the final number of eligible dishes was 1361. We were able to extract kcal information for 1009 dishes (74.1% of eligible items) and the remaining dishes were treated as missing data and not included in analyses. The missing information for the 25.9% of the items was due to lack of nutritional information provided by restaurants. See table 1 for number of eligible dishes per restaurant.

### Mean kcal content of menu items

For all three groups (starters, sides and desserts) two-level model (dishes within restaurants) was a better fit of the data than a single level model. The variance partition coefficient; the total residual variance which is attributable to restaurants rather than individual dishes was 14.7% (model fit:$\chi^2$ (1) = 18.1, $p < .001$) for starters, 13.8% (model fit: $\chi^2$ (1) = 35.0, $p <.001$) for sides, and 45.0% (model fit: $\chi^2$ (1) = 197.5, $p < .001$) for desserts, indicating that multi-level modelling was appropriate. In a one-level model (for the descriptive purposes), the average number of kcals for starters was 488.0 (SE=15.6), for sides was 397.5 (SE=14.9) and for desserts was 430.6 (SE=11.5).

Next, we used a two-level model to compare the average number of kcal in sides and in desserts between fast-food and full-service restaurants, as there were no starters identified in the fast-food restaurants. Type of restaurant (fast food vs full service) was not a significant predictor of kcal content for sides (β=0.1, SE=2.8 (95% CIs −5.5 to 5.6), p=0.49) indicating that sides offered at fast-food restaurants had on average only 0.1 kcal more energy than sides from fast-food restaurants. Desserts had on average 241.2 more kcal in full service than in fast food chains (β=241.2,

**Table 1** Kilocalorie content of dishes from eligible restaurant chains included in analyses

| Restaurant type | Restaurant chain name | N | Starters | | Sides | | Desserts | |
|---|---|---|---|---|---|---|---|---|
| | | | Number | Mean (SD) kcals | Number | Mean (SD) kcal | Number | Mean (SD) kcals |
| Fast-food | Burger King | 39 | – | N/A | 19 | 332.1 (136.7) | 20 | 311.5 (153.4) |
| | KFC | 83 | – | N/A | 73 | 562.9 (240.1) | 10 | 309.5 (81.1) |
| | Leon | 25 | – | N/A | 10 | 209.3 (65.0) | 15 | 270.0 (81.0) |
| | McDonalds | 23 | – | N/A | 6 | 216.3 (167.1) | 17 | 242.0 (115.5) |
| | Subway | 5 | – | N/A | 1 | 705.0 (–) | 4 | 213.8 (2.2) |
| | Wimpy | 20 | – | N/A | 12 | 282.2 (168.8) | 8 | 456.3 (230.2) |
| | All fast-food restaurants (n=6)* | 195 | – | N/A | 121 | 453.6 (249.7) | 74 | 297.2 (142.4) |
| | n (%)>600 kcal† | | –(N/A) | | 40 (33.1) | | 3 (4.1) | |
| Full-service | All bar one | 11 | – | N/A | 6 | 447.7 (140.0) | 5 | 587.2 (222.6) |
| | Ask | 89 | 27 | 565.7 (278.0) | 9 | 315.2 (336.1) | 53 | 273.2 (137.3) |
| | Bills | 33 | 19 | 318.9 (123.4) | 5 | 265.4 (161.7) | 9 | 535.4 (298.2) |
| | Chef and Brewer | 39 | 8 | 481.1 (124.8) | 13 | 302.6 (213.4) | 18 | 486.1 (300.3) |
| | Ember Inns | 29 | 7 | 307.6 (102.9) | 8 | 206.1 (135.2) | 14 | 522.2 (135.1) |
| | Flaming Grill | 26 | 4 | 644.8 (111.4) | 14 | 459.4 (233.9) | 8 | 767.1 (343.7) |
| | Harvester | 34 | 14 | 424.5 (119.2) | 9 | 254.0 (148.7) | 11 | 670.9 (156.2) |
| | Hungry horse | 44 | 18 | 660.2 (247.5) | 16 | 454.3 (280.0) | 10 | 867.9 (517.6) |
| | JD Wetherspoon | 24 | – | N/A | 14 | 406.1 (325.9) | 10 | 571.3 (169.3) |
| | Nando's | 40 | 6 | 486.0 (265.5) | 24 | 365.0 (320.7) | 10 | 330.0 (217.4) |
| | Old English Inn | 87 | 11 | 433.5 (199.6) | 10 | 364.4 (209.4) | 66 | 408.3 (141.5) |
| | Pizza Express | 49 | 11 | 379.5 (196.4) | 3 | 328.7 (126.0) | 35 | 467.8 (97.9) |
| | Pizza Hut | 18 | 11 | 463.6 (107.7) | 4 | 412.5 (158.2) | 3 | 624.7 (82.7) |
| | Sizzling Pubs | 36 | 13 | 477.7 (167.0) | 10 | 391.6 (227.4) | 13 | 723.5 (210.8) |
| | Slug and Lettuce | 17 | – | N/A | 7 | 754 (656.4) | 9 | 400.9 (147.7) |
| | Stone house | 33 | 18 | 622.6 (329.9) | 2 | 88.0 (26.9) | 13 | 686.4 (230.0) |
| | Table Table | 30 | 9 | 455.9 (154.4) | 12 | 303.3 (159.7) | 9 | 519.3 (223.1) |
| | Toby's Carvery | 33 | 10 | 423.3 (134.3) | – | N/A | 23 | 671.5 (251.2) |
| | Vintage Inns | 27 | 9 | 357.7 (271.9) | 8 | 239.5 (206.7) | 10 | 738.9 (345.9) |
| | Wagamama | 35 | – | N/A | 22 | 328.1 (117.6) | 13 | 352.6 (120.4) |
| | Zizzi | 81 | 17 | 575.6 (155.1) | 1 | 222.0 (*-) | 63 | 246.5 (150.0) |
| | All full-service restaurants (n=21)* | 815 | 212 | 488.0 (227.7) | 197 | 397.5 (265.3) | 405 | 430.6 (251.5) |
| | n (%)>600 kcal*† | | 56 (26.4) | | 29 (14.7) | | 95 (23.5) | |

– indicates the absence of dish from restaurant chain menu.(–) indicates the absence of SD as only one eligible dish from restaurant.

N/A - Non applicable

*For descriptive purposes, values in this row represent the one-level mean (SD) of individual restaurant values for mean kcals per dish.

†The values presented in these rows are the numbers of the dishes exceeding the 600 kcal and their representation among the total meals identified (n (%)).

SE=65.4 (95% CIs 113.0 to 369.4), p=0.001) and this difference was statistically significant.

### Mean kcal content of specific dish types

The most common side available was chips/fries. To compare the average kcal content of chips between the fast-food and full-service restaurants, we selected only chips/fries menu options that were made of potato, plain, with no sauces or spices, toppings or extras and were served as sides. The inclusion criteria resulted in 40 eligible menu items, offered in 19 restaurants (out of the 27), including 5 out of 6 fast-food chains (n=13 items), and 14 out of 21 full-service chains (n=27 items). The small number of items eligible for this subanalysis did not lend itself to multilevel analysis so we used Welch's t-test

to compare the types of restaurants. The average number of kcals was 441.9 (SE=33.1) across all restaurants. Chips/fries in full-service restaurants had on average 197.0 kcal more than in fast-food restaurants (505.9 kcal vs 308.9 kcal) and this difference was statistically significant (t (38)=3.9, p<0.01, d=1.2). Ice cream dishes were the most frequently served dessert across restaurants. We selected only ice creams made of dairy cream; therefore, items such as sorbets, vegan ice creams (or combinations of flavours including either of these) and other desserts that included ice cream (such as cake with ice cream) were excluded from the comparison. Ice creams were served in 19 out of 27 restaurants (4 of 6 fast foods and 15 of 21 full service), with a total of 114 items (24 in fast foods and 90 in full service). The average amount of kcals in ice cream dishes was 389.2 (SE=27.1). Full-service restaurant ice cream dishes had on average 190.9 kcal more than fast-food ice cream dishes (429.4 kcal vs 238.5 kcal) and this difference was statistically significant (t (85)=4.5, p<0.001, d=0.8). See table 2 for number of eligible specific dish types per restaurant.

### Menu items >600 kcal

Of the 212 starters identified, 56 (26.4%) exceeded 600 kcal per dish and all starters were from full-service restaurants. Of the 318 sides, 69 (21.7 %) had >600 kcal. A multilevel logistic regression model demonstrated that the proportion of sides >600 kcal was not significantly larger in fast-food restaurants compared to full-service restaurants (Wald statistic (1) = 4.32, p = 0.04 OR = 1.52 (95% CIs 0.14 to 16.10), p =0.48). Among the 479 identified desserts, 98 (20.5 %) exceeded 600 kcal. A multilevel logistic regression model demonstrated that the proportion of desserts exceeding the 600 kcal was 14 times larger in full service compared with fast food restaurants (Wald statistic (1)=7.7, p<0.01; OR=14.01 (95% CIs 1.95 to 101.49), p<0.01).

### DISCUSSION

The present study examined the energy content of starters, sides and desserts sold by major UK restaurant chains. We found that the average number of kcals in starters, sides and desserts was 488.0 (SE=15.6), 397.5 (SE=14.9) and 430.6 (SE=11.5) kcal, respectively. We also examined the proportion of these dishes that we deemed to be 'excessive' by identifying those with more than 600 kcal; the recommended kcal content of a full lunch or dinner meal in the UK.[19] We identified that one in four starters and one in five sides and desserts exceeded the amount of energy recommended for a full meal. Results also indicated that kcal content of dishes was associated with restaurant type. When comparing types of restaurants, we found that desserts were significantly higher in kcals in full-service versus fast-food restaurants.

Our results are in line with studies that have examined the energy content of North American restaurant food and a recent UK study finding an excessive number of kcals in

**Table 2** Kilocalorie content of chips/fries and ice cream dishes from eligible restaurant chains included in analyses

| Restaurant chain | N | Chips/fries Mean kcal (SD) | N | Ice cream Mean kcal (SD) |
|---|---|---|---|---|
| Burger King | 4 | 342.5 (113.2) | 6 | 168.3 (80.6) |
| KFC | 2 | 372.5 (95.5) | 4 | 253.8 (94.2) |
| Leon | 2 | 174.0 (–) | – | N/A |
| McDonalds | 3 | 339.3 (103.5) | 10 | 211.0 (76.6) |
| Subway | – | N/A | – | N/A |
| Wimpy | 2 | 267.5 (0.7) | 4 | 397.3 (252.5) |
| All fast-food restaurants* | 13 | 308.9 (101.7) | 24 | 238.5 (138.6) |
| All bar one | 2 | 452.0 (75.0) | – | N/A |
| Ask | – | N/A | 16 | 272.8 (72.0) |
| Bills | 2 | 429.5 (113.8) | 1 | 107.0 (–) |
| Chef and Brewer | 3 | 478.7 (133.5) | 1 | 951.0 (–) |
| Ember Inns | – | N/A | 1 | 338.0 (–) |
| Flaming Grill | 1 | 546.0 (–) | 1 | 1421.0 (–) |
| Harvester | 2 | 469.5 (47.4) | – | N/A |
| Hungry horse | 3 | 530.0 (112.9) | 3 | 1223.7 (903) |
| JD Wetherspoon | 1 | 955.0 (–) | – | N/A |
| Nando's | 3 | 680.3 (503.8) | 4 | 140.5 (28.5) |
| Old English Inn | 1 | 764.0 (–) | 12 | 498.5 (197.1) |
| Pizza Express | – | N/A | 11 | 481.1 (63.4) |
| Pizza Hut | 2 | 493.5 (195.9) | – | N/A |
| Sizzling Pubs | 1 | 503.0 (–) | 5 | 688.8 (286) |
| Slug and Lettuce | – | N/A | – | N/A |
| Stone house | 1 | 107.0 (–) | 1 | 800.0 (–) |
| Table Table | 3 | 347.3 (17.2) | 2 | 596.5 (9.2) |
| Toby's Carvery | – | N/A | 4 | 559.0 (147.2) |
| Vintage Inns | 2 | 493.5 (13.4) | – | N/A |
| Wagamama | – | N/A | 5 | 412.0 (97.7) |
| Zizzi | – | N/A | 23 | 270.5 (146.7) |
| All full-service restaurants* | 27 | 505.9 (219.2) | 90 | 429.4 (289.4) |

– indicates the absence of dish from restaurant chain menu. (–) indicates the absence of SD as only one eligible dish from restaurant. N/A - Non applicable
*For descriptive purposes, values in this row represent the one-level mean (SD) of individual restaurant values for mean chips/fries dishes kcals and mean ice creams dishes kcals.

menu items in the eating-out sector and a general trend for full-service restaurant menu items to on average be more calorific than fast-food restaurants.[10 12 14 20–22] Based on the results from our current study and the previous UK study of main meals,[12] the average energy content of a three-course meal (starter, main meal, dessert, without the addition of an extra side) in a major chain restaurant in the UK would be approximately 1896 kcal, which equates to over three times the recommended energy intake for a main meal, and 95% of the recommended daily consumption of kcals for women or 76% for men.

Although individual energy requirements vary according to levels of physical activity, age, gender and body mass, frequent eating out of home combined with the relatively high energy content of restaurant dishes (including starters, sides and desserts) may contribute to excessive energy intake that is now common in the UK and other high-income countries. The present research has relevance to public health policy and our results also suggest that policy actions which result in the reduction of the energy content of restaurant food are urgently needed. In September 2018, the UK government launched an open consultation on kcal labelling for food and drink served outside of the home. As our study shows, starters, sides and desserts can be highly calorific and, in some cases, exceed the amount of energy recommended for a single meal. A recently published study performed two meta-analyses to study the effect of menu energy labelling on consumer choice and the energy content of menu items. It showed that there was a reduction in kcals ordered by consumers and a reduction in energy content of menu items provided by restaurants when the energy content of meals was displayed at the point of choice.[23] Thus, this research supports the proposition that menu labelling may benefit public health through two main channels: industry reformulation and individual behaviour change.[24] Given that kcal labelling is only likely to have a small effect on daily energy intake, a combination of this and other population-wide interventions will be required to improve diet and reduce obesity.

We also found a high degree of variability in kcal content in similar dishes across restaurants, which may make it difficult for consumers to estimate energy content without access to nutritional information. For example, the most calorific portion of chips/fries offered in studied restaurant chains had nearly 12 times more than the least calorific (107 and 1256 kcal) and it was common for ice cream desserts to vary dramatically in kcal content. Although due to the methodological challenges, we did not include smaller chains or independent restaurants in our study, evidence from US studies suggests that both chain and non-chain restaurants tend to serve highly calorific foods.[10] Therefore, mandatory kcal labelling in the UK out-of-home food sector would be appropriate. The present study is the first of which we are aware of that assesses the energy content of starters, sides and desserts in the UK eating out sector, and the results may be of use to future evaluations of how the out-of-home food sector respond to voluntary or mandatory public health actions through food product reformulation.

### Limitations

As there are no guidelines for the limits of calories in different courses of the meal, we examined the proportion of meals exceeding Public Health England's recommendation of 600 kcal or less per entire meal (lunch or dinner). A further limitation of the study was that we were only able to make use of nutrition data from restaurants that made this information available, which excluded several dishes and restaurants from our analyses. This presents a potential source of bias if the kcal content of restaurants that do not provide nutrition information differs to that of restaurants that do. It is also possible that restaurant-provided nutrition information is inaccurate, although research suggests that any inaccuracy may be relatively small.[25]

## CONCLUSIONS

The energy content of sides, starters and desserts sold in major UK restaurants is high. One in four starters and one in five sides and desserts in UK chain restaurants exceed the recommended energy intake for an entire meal.

**Contributors** ER and MM designed the study. MM, CR and FS contributed to data collection. AH provided an advice and expertise at all stages and helped solving eligibility disagreements. AJ and MM were responsible for data analysis. MM was responsible for initial drafting of the paper and all authors approved the manuscript and had full access to the data.

**Funding** The MRC (MR/N00218/1) part fund ER's salary.

**Disclaimer** The views expressed in this publication are those of the authors and not necessarily those of the MRC.

**Competing interests** All authors have completed the ICMJE uniform disclosure form at www.icmje.org/coi_disclosure.pdf. ER has been a named investigator on research projects funded by the American Beverage Association but does not consider this funding a conflict of interest.

**Patient consent for publication** Not required.

**Provenance and peer review** Not commissioned; externally peer reviewed.

**Data availability statement** Data are available in a public, open access repository.

**ORCID iD**
Magdalena Muc http://orcid.org/0000-0001-6323-9973

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
