## [Reviewer comments · BMJ Open]

ARTICLE DETAILS

TITLE (PROVISIONAL)	A bit or a lot on the side? An observational study of the energy content of starters, sides and desserts in major UK restaurant chains
AUTHORS	Muc, Magdalena; Jones, Andrew; Roberts, Carl; Sheen, Florence; Haynes, Ashleigh; Robinson, Eric

VERSION 1 – REVIEW

REVIEWER	Lacey A. McCormack South Dakota State University, USA
REVIEW RETURNED	06-Mar-2019

GENERAL COMMENTS	Most of the comments have been added to the pdf of the manuscript, and a few have been noted below.  -Recommend person-first language for overweight/obesity -- i.e. '... classed as having obesity.' -Recommend saying 'outside of the home' or 'away from home' when talking about food not prepared at home. -Recommend revising the 'Limitations' section for clarity and to address limitations with study design only. The reviewer provided a marked copy with additional comments. Please contact the publisher for full details.
--

REVIEWER	Dr. Caroline Glagola Dunn, PhD, RD Harvard T H Chan School of Public Health, Department of Health Policy and Management
REVIEW RETURNED	07-May-2019

GENERAL COMMENTS	Reviewer: Dr. Caroline Glagola Dunn, PhD, RD Research Associate Harvard T H Chan School of Public Health Department of Health Policy and Management General comments: This paper describes an assessment of the kilocalorie content of side dishes, appetizers, and desserts at 27 UK fast food or fast casual chain restaurants. This is an important area of research, and the paper adds to the current literature by examining menu items in UK restaurants that have not been widely studied. Generally, the paper is well written, but would benefit from the additional details in the methods section and an expansion of the discussion. Thank you for the opportunity to review the paper. Overall, I believe this paper would benefit from additional edits.
---

Abstract: The authors do not currently include any statistical approaches in their methods, which are a large focus of the methods section of the paper. The abstract could be edited (consider the first sentence of the Objectives section, as it provides background and not an objective; consolidate the conclusions to one sentence) to provide additional room to describe the statistical approaches.

Strengths and Limitations of the Study:

Point 3 (lines 53-54): If you only included restaurants with 50 or more outlets in the UK (abstract), how do you then make a statement about non-chain restaurants here? You state that both chain and non-chain restaurants tend to serve highly calorific foods – specify that this is a finding from previous research and not from your study. Also, please use consistent punctuation; point three does not have a period at the end of the sentence.

Introduction

Line #	Comment
--	The introduction is well written, and provides justification (citations needed, see below) for the current study. If the authors need to reduce word count in order to expand on other ideas in the article (specifically the discussion), information in the introduction could be consolidated.
81	The authors state that patrons regularly order starters, sides and/or desserts as part of their meal out of the home. This statement is a critical part of the justification for the importance of your study; please provide a reference for this statement.
General	Please use consistent language to describe starters, sides, and desserts (e.g., sides vs side dishes vs side items) throughout the paper. Establish this nomenclature in the introduction.

Methods

Line #	Comment
95-102	The authors define fast-food, but do not offer a definition for full-service restaurants. Please provide a definition. If the authors have not previously defined this, suggested references are offered below: Schoffman DE, Davidson CR, Hales SB, Crimarco AE, Dahl AA, Turner-McGrievy GM. The fast-casual conundrum: Fast-casual restaurant entrees are higher in calories than fast food. Journal of the Academy of Nutrition and Dietetics . 2016 Oct 1;116(10):1606-12. Jarlenski MP, Wolfson JA, Bleich SN. Macronutrient composition of menu offerings in fast food restaurants in the US. American journal of preventive medicine . 2016;51(4):e91-e97. Moran AJ, Block JP, Goshev SG, Bleich SN, Roberto CA. Trends in nutrient content of children’s menu items in US chain restaurants. American journal of preventive medicine . 2017;52(3):284-291.
103-108	Restaurants’ online menus may detect a user’s location (postal code) and present regionally specific

		menu items. Did the authors encounter this issue, or were they required to enter their location to access any of the restaurant menus? If so, how did they proceed?
109-130		The authors do a nice job of describing their inclusion/exclusion criteria for starters, sides, and dessert items. Do the authors have any record of the number for each of these exclusions? It would be helpful for the reader to know the scope of the analysis (e.g., were the authors able to evaluate the majority of sides/starters/desserts available on selected menus?).
125		You appear to be missing a period at the end of the sentence "...chocolate cake with custard")"
128-130		This sentence is confusing to the reader; please rephrase without using and/or.
135		Please fix citation style for reference (12)
139		In the abstract, the authors state that data extraction occurred in October to November 2018. Here the authors state that extraction occurred in October 2018. Please clarify.
Statistical Analysis		
Line #	Comment	
142-154	The authors have done a nice job of describing their process for assessing the appropriateness of their modeling approach. It would be helpful to the reader if the authors included a list of the variables for each model level (i.e., how the variables were nested) and identified any random or fixed effects. You provide some of this information in the results, but it would be beneficial for the reader to have this in the methods.	
162-165	Please provide additional information on how items were identified, categorized, coded by the research team. This appears to be an important part of your paper, but the methods do not adequately describe what was done. This information may also fit better in the section that currently ends on line 140 (prior to statistical analysis)	
Results		
Line #	Comment	
180-181	Please identify the reason(s) that you were not able to extract data for 25.9% of the sample.	
192	Be consistent with the number of decimal places you report (for starters and sides, you report SE out to one decimal; for desserts, you report SE out to two).	
199, 225	Check your use of parentheses and be sure to close all brackets or use a semicolon.	
201	The mean kcal content of specific dish types seems to be an important part of your results. However, you do not provide robust methods for identifying/classifying dish type in the methods (162-165). If this is, in fact, an important aspect of your results. Please provide more information on how "type" was identified, classified, and coded.	
299	Please use semicolon to separate your OR and CIs instead of a colon.	

234-235	Here you refer to stater dishes, side dishes, and dessert dishes. Earlier, you use the terms starters, sides, and deserts. Please be consistent with your terminology throughout.
Discussion	
Line #	Comment
--	In general, the discussion is short and does not provide adequate comparison to previous literature or implications for public health policy or individual behavior change. If the authors are tight on word count, the introduction could be condensed to create room for a more robust discussion section.
244-245	In what ways do your results align with previous studies? Please expand on this.
245	Reference 20 is a controlled field experiment on menu labeling. It does not appear to be examining the energy content of restaurant foods directly and does not appear to be similar to the current study. Consider recent publications listed below that may offer more appropriate comparisons: Schoffman DE, Davidson CR, Hales SB, Crimarco AE, Dahl AA, Turner-McGrievy GM. The fast-casual conundrum: Fast-casual restaurant entrees are higher in calories than fast food. Journal of the Academy of Nutrition and Dietetics. 2016 Oct 1;116(10):1606-12. Bleich SN, Wolfson JA, Jarlenski MP. Calorie changes in large chain restaurants: declines in new menu items but room for improvement. American journal of preventive medicine. 2016 Jan 1;50(1):e1-8. Bleich SN, Wolfson JA, Jarlenski MP, Block JP. Restaurants with calories displayed on menus had lower calorie counts compared to restaurants without such labels. Health affairs. 2015 Nov 1;34(11):1877-84.
246	Does this “whole meal” (starter, main dish, dessert) include a side dish? If so, please specify; if not, please include and adjust the values.
259-265	The authors make a good point about variability being a potential cause of confusion for consumers and identifying access to nutrition information as a potential solution. The authors also offer a relevant policy solution. However, research shows that individual behavior is difficult to change (i.e., consumers tend to stick to the status quo). In addition, one of their references (currently 20) states that the provision of calorie information on menus reduces calories by about 10 kcal/day, which means a three-year period is required to reduce an individual’s weight by 1 pound. Could the authors offer potential solutions that do not rely on individual behavior change?
276	The term “these and the results” is awkward and feels like an incorrect edit. Please revise.

	268-269	This is the first time that the authors bring up reformulation. If this is a potential solution (see my comment above for lines 259-265), the authors should discuss it in more depth.
Tables: Footnotes for Table 1 and Table 2 do not include a superscript a, only b. Is this correct, or should there be a superscript a? The row structure of Table 1 is also confusing, it appears that several of the rows in either fast-food or full-service should have been merged in the restaurant type column.		

VERSION 1 – AUTHOR RESPONSE

Reviewer: 1

Reviewer Name: Lacey A. McCormack

Institution and Country: South Dakota State University, USA

Reviewer's comment on page 4 lines 59 changing the verb "being" to "having":

In line 57 we have changed the word "being" to "having" as suggested.

Reviewer's comment on page 4 lines 61 suggests changing the verb "has" to "have".

In line 59 we have changed the verb "has" to "have".

Reviewer's comment on page 4 lines 61 and 67: recommend changing these to 'outside of home' or 'away from home' to keep consistent with literature

As for the use of the phrase "Eating out" we searched the pub-med database and "eating out" is the most commonly used phrase in published research to define eating in restaurants/out of home. We therefore have retained this terminology in the manuscript.

Reviewer's comment on page 6 line 129: is this supposed to say '...available all year round and/or sold everyday..'?

We have revised the lines 129-131 and the sentence has been changed and now states: "Finally, to minimize effects of season, we only included options that were available all year round and sold everyday (e.g. we excluded dishes sold only on specific days, such as 'soup of the day')."

Reviewer's comment on page 7 line 135: format all references the same

We have changed the format of the reference in line 140.

Reviewer's comment on page 11 line 235: should include SE here as well

We have added the SE values in the 243 line in the discussion.

Reviewer's comment on page 12 line 273: I don't know that this is a limitation with study design per se, if chain locations were outlined in the eligibility criteria -- it is likely more of a discussion point.

We have moved the sentence referring to the small chains and independent restaurants from the limitations and moved up to lines 285-287 and it now states: "Although due to the methodological challenges, we did not include smaller chains or independent restaurants in our study, evidence from US studies suggests that both chain and non-chain restaurants tend to serve highly calorific foods."

Reviewer's comment on page 13 line 279: and many locations

The sentence in lines 297-299 now states "A further limitation of the study was that we were only able to make use of nutrition data from restaurants that made this information available, which excluded several dishes and restaurants from our analyses."

-Recommend person-first language for overweight/obesity -- i.e. '... classed as having obesity.'
We agree and we have revised the language accordingly.

-Recommend saying 'outside of the home' or 'away from home' when talking about food not prepared at home.

As above, we consulted the literature and the dictionary and decided to use "Eating out" to refer to eating out of home, in restaurants.

-Recommend revising the 'Limitations' section for clarity and to address limitations with study design only.

As above, we have revised the section accordingly.

Reviewer 2:

Dr. Caroline Glagola Dunn, PhD, RD

Research Associate

Harvard T H Chan School of Public Health

Department of Health Policy and Management

Abstract:

The authors do not currently include any statistical approaches in their methods, which are a large focus of the methods section of the paper. The abstract could be edited (consider the first sentence of the Objectives section, as it provides background and not an objective; consolidate the conclusions to one sentence) to provide additional room to describe the statistical approaches.

As advised, we have removed the first sentence in the objectives section.

We have also added information on the statistical approach adopted (page 2, lines 31-32): "We used multilevel modelling to examine whether mean kcal content of in dishes differed in fast-food vs. full-service restaurants."

Strengths and Limitations of the Study:

Point 3 (lines 53-54): If you only included restaurants with 50 or more outlets in the UK (abstract), how do you then make a statement about non-chain restaurants here? You state that both chain and nonchain restaurants tend to serve highly calorific foods – specify that this is a finding from previous research and not from your study. Also, please use consistent punctuation; point three does not have a period at the end of the sentence.

As requested, we have now changed this section to: "Smaller chains and independent restaurants were not included, however literature shows that both chain and non-chain restaurants tend to serve highly calorific foods." See page 3, line 51-52.

Introduction

-- The introduction is well written, and provides justification (citations needed, see below) for the current study. If the authors need to reduce word count in order to expand on other ideas in the article (specifically the discussion), information in the introduction could be consolidated.

As the word limit was not exceeded and we considered the introduction to contain important content we did not consolidate the introduction.

81 The authors state that patrons regularly order starters, sides and/or desserts as part of their meal out of the home. This statement is a critical part of the justification for the importance of your study; please provide a reference for this statement.

Since we are not aware of any study that examines how many courses people usually order, we have changed this statement:

“However, previous research has focused on main meals and consumers eating out are offered a choice of starters, sides and/or desserts on restaurant menus.” See pages 4-5, lines 78-80.

General Please use consistent language to describe starters, sides, and desserts (e.g., sides vs side dishes vs side items) throughout the paper. Establish this nomenclature in the introduction.

We have changed terminology throughout the manuscript to ensure consistency: “starters, sides and desserts” as in the title.

Methods

95-102 The authors define fast-food, but do not offer a definition for full-service restaurants.

We have added a definition and citation: “We classified full-service chains as restaurants where consumers primarily order and are served while seated at a table 19.” See page 5, line 86-87.

103-108 Restaurants’ online menus may detect a user’s location (postal code) and present regionally specific menu items. Did the authors encounter this issue, or were they required to enter their location to access any of the restaurant menus? If so, how did they proceed?

We have now clarified this point: “If a specific geographical location was required to access a restaurant chain menu we chose London (largest city in the UK).” See page 6, line 105-106.

109-130 The authors do a nice job of describing their inclusion/exclusion criteria for starters, sides, and dessert items. Do the authors have any record of the number for each of these exclusions? It would be helpful for the reader to know the scope of the analysis (e.g., were the authors able to evaluate the majority of sides/starters/desserts available on selected menus?).

We do not unfortunately. The approach used is explained on page 7, line 143-148. It is however important to note that each reviewer assessed those sections of every menu, in addition to examining whether dishes that could be considered to be sides/starters/desserts appeared anywhere else on the menu.

125 You appear to be missing a period at the end of the sentence “...chocolate cake with custard)”
We have added a period.

128-130 This sentence is confusing to the reader; please rephrase without using and/or.

We have rephrased the sentence: “Finally, to minimize effects of season, we only included options that were available all year round and sold everyday (e.g. excluded dishes sold only on specific days, such as ‘soup of the day’).” See pages 6-7, lines 129-131.

135 Please fix citation style for reference (12)

Correction made. See page 7, line 140.

139 In the abstract, the authors state that data extraction occurred in October to November 2018.

Here the authors state that extraction occurred in October 2018. Please clarify.

In October we accessed menus, in November we accessed nutritional files. This is now specified in the text. See page 7, lines 134 and 150.

Statistical Analysis

142-154 The authors have done a nice job of describing their process for assessing the appropriateness of their modelling approach. It would be helpful to the reader if the authors included a

list of the variables for each model level (i.e., how the variables were nested) and identified any random or fixed effects. You provide some of this information in the results, but it would be beneficial for the reader to have this in the methods.

We have provided more information: "Menu items were nested within individual restaurants so we planned to use multi-level analyses (levels: menu item, restaurant) with random intercept at the restaurant level and fixed slopes." See page 7-8, line 153-155.

162-165 Please provide additional information on how items were identified, categorized, coded by the research team. This appears to be an important part of your paper, but the methods do not adequately describe what was done. This information may also fit better in the section that currently ends on line 140 (prior to statistical analysis).

We have now provided more detail. See page 7, lines 143-148.

"Because variability in menu item kcal content between restaurant types may be in part explained by the two types of restaurant serving different types of dishes we examined whether there were dishes that were routinely sold by both types of restaurant (e.g. side of fries/chips, salad) and compared the average number of kcals for these dishes by restaurant type. Since the names of the same dishes could vary between menus, coding of these items was completed by one researcher and cross-checked by a second researcher.

Results

180-181 Please identify the reason(s) that you were not able to extract data for 25.9% of the sample. We have added the information as follows: "The missing information for the 25.9% of the items was due to lack of nutritional information provided by the restaurant." See page 9, lines 191-192.

192 Be consistent with the number of decimal places you report (for starters and sides, you report SE out to one decimal; for desserts, you report SE out to two).

We have reduced the number of digits to one in all but the OR results, which habitually require 2 decimal places.

199, 225 Check your use of parentheses and be sure to close all brackets or use a semicolon. We have added missing parentheses to the manuscript.

201 The mean kcal content of specific dish types seems to be an important part of your results. However, you do not provide robust methods for identifying/classifying dish type in the methods (162-165). If this is, in fact, an important aspect of your results. Please provide more information on how "type" was identified, classified, and coded.

We have now added this information to the manuscript. See page 7, lines 143-151. "Because variability in menu item kcal content between restaurant types may be in part explained by the two types of restaurant serving different types of dishes we examined whether there were dishes that were routinely sold by both types of restaurant (e.g. side of fries/chips, salad) and compared the average number of kcals for these dishes by restaurant type. Since the names of the same dishes could vary between menus, coding of these items was completed by one researcher and cross-checked by a second researcher.

Extraction of dish kcal content. Two researchers accessed the online nutritional information for each restaurant (November 2018) and extracted the number of kcals per menu item. A third researcher independently cross-checked kcal extraction for accuracy."

299 Please use semicolon to separate your OR and CIs instead of a colon. We have used a semicolon, as suggested.

234-235 Here you refer to starter dishes, side dishes, and dessert dishes. Earlier, you use the terms starters, sides, and deserts. Please be consistent with your terminology throughout. We now use consistent terminology throughout: “starters, sides and desserts”.

Discussion

244-245 In what ways do your results align with previous studies? Please expand on this. We now discuss this. See page 11-12, line 252-255.

245 Reference 20 is a controlled field experiment on menu labeling. It does not appear to be examining the energy content of restaurant foods directly and does not appear to be similar to the current study. Consider recent publications listed below that may offer more appropriate comparisons: Schoffman DE, Davidson CR, Hales SB, Crimarco AE, Dahl AA, Turner-McGrievy GM. The fast-casual conundrum: Fast-casual restaurant entrees are higher in calories than fast food. *Journal of the Academy of Nutrition and Dietetics*. 2016 Oct 1;116(10):1606-12.

Bleich SN, Wolfson JA, Jarlenski MP. Calorie changes in large chain restaurants: declines in new menu items but room for improvement. *American journal of preventive medicine*. 2016 Jan 1;50(1):e1-8.

Bleich SN, Wolfson JA, Jarlenski MP, Block JP. Restaurants with calories displayed on menus had lower calorie counts compared to restaurants without such labels. *Health affairs*. 2015 Nov 1;34(11):1877-84.

We have replaced reference 20 with:

Bleich SN, Wolfson JA, Jarlenski MP, Block JP. Restaurants with calories displayed on menus had lower calorie counts compared to restaurants without such labels. *Health affairs*. 2015 Nov 1;34(11):1877-84.

246 Does this “whole meal” (starter, main dish, dessert) include a side dish? If so, please specify; if not, please include and adjust the values.

We have now specified this. See page 12, line 257.

259-265 The authors make a good point about variability being a potential cause of confusion for consumers and identifying access to nutrition information as a potential solution. The authors also offer a relevant policy solution. However, research shows that individual behavior is difficult to change (i.e., consumers tend to stick to the status quo). In addition, one of their references (currently 20) states that the provision of calorie information on menus reduces calories by about 10 kcal/day, which means a three-year period is required to reduce an individual’s weight by 1 pound. Could the authors offer potential solutions that do not rely on individual behavior change?

We now discuss the public health relevance of energy labelling and consider how it may improve diet beyond individual behaviour change. See page X, line X. See page 12, line 270-278.

276 The term “these and the results” is awkward and feels like an incorrect edit. Please revise.

We have revised this terminology. See page 13, line 290.

268-269 This is the first time that the authors bring up reformulation. If this is a potential solution (see my comment above for lines 259-265), the authors should discuss it in more depth.

We discussed in more depth the potential effect of the menu labelling on reformulation. See page 12, line 270-278.

Tables: Footnotes for Table 1 and Table 2 do not include a superscript a, only b. Is this correct, or should there be a superscript a? The row structure of Table 1 is also confusing, it appears that several of the rows in either fast-food or full-service should have been merged in the restaurant type column.

We have revised and corrected the superscript and merged the column to refer to “fast food” and “full-service” more clearly.

VERSION 2 – REVIEW

REVIEWER	Dr. Caroline Glagola Dunn, PhD, RD Harvard T H Chan School of Public Health, Department of Health Policy and Management; Boston, MA, USA
REVIEW RETURNED	23-Aug-2019

GENERAL COMMENTS	Thank you for the opportunity to review this re-submission. Based on the updated document and response to reviewers, I recommend accepting with minor suggestions (points below), and do not feel the need to review the manuscript again, if revisions are approved by the editor. Abstract: The abstract is well-written and clearly communicates the objectives, methods, findings, and implications. Thank you for updating the methods to provide additional information to the reader about the models used in this analysis. Introduction: Thank you for addressing my previous questions/suggestions. Again, the introduction is well written, understandable, and justifies the analysis. Methods: Your additional methodological detail is appreciated. Thank you for making the updates and providing additional detail about the process of reviewing and coding items, specifically. Lines 175-176: Correct subject verb agreement issue (i.e., “a” is singular, “regressions” is plural) Line 149: “Since” indicates a passage of time, please change to “because” Results: I appreciate the additional text that outlines how your findings align with current research. I also appreciate the additional discussion of potential policy implications of menu labeling outside of a reliance on individual behavior change. Overall: Thank you, again, for your revisions and responses to the original critique. With minor edits, I am comfortable recommending this manuscript for publication.
---